# Dynamics of Cytokine, SARS-CoV-2-Specific IgG, and Neutralizing Antibody Levels in COVID-19 Patients Treated with Convalescent Plasma

**DOI:** 10.3390/diseases11030112

**Published:** 2023-08-30

**Authors:** Pornpitra Pratedrat, Duangnapa Intharasongkroh, Jira Chansaenroj, Preeyaporn Vichaiwattana, Donchida Srimuan, Thaksaporn Thatsanatorn, Sirapa Klinfueng, Pornjarim Nilyanimit, Chintana Chirathaworn, Pawinee Kupatawintu, Dootchai Chaiwanichsiri, Nasamon Wanlapakorn, Yong Poovorawan

**Affiliations:** 1Center of Excellence in Clinical Virology, Faculty of Medicine, Chulalongkorn University, Bangkok 10330, Thailand; pornpitra@nmu.ac.th (P.P.); job151@hotmail.com (J.C.); preeyaporn.vic@chulahospital.org (P.V.); donchida.s@gmail.com (D.S.); thaksapohnl@hotmail.com (T.T.); sirapa.klinfueng@gmail.com (S.K.); mim_bhni@hotmail.com (P.N.); nasamon.w@chula.ac.th (N.W.); 2Department of Basic Medical Science, Faculty of Medicine Vajira Hospital, Navamindradhiraj University, Bangkok 10300, Thailand; 3National Blood Centre, Thai Red Cross Society, Bangkok 10330, Thailand; duangnapa.i@redcross.or.th (D.I.); pawinee.k@redcross.or.th (P.K.); dootchai.c@redcross.or.th (D.C.); 4Department of Microbiology, Faculty of Medicine, Chulalongkorn University, Bangkok 10330, Thailand; chinchula99@gmail.com; 5Royal Society of Thailand (FRS(T)), Sanam Sueapa, Dusit, Bangkok 10330, Thailand

**Keywords:** retrospective study, convalescent plasma treatment, COVID-19, cytokine, antibodies

## Abstract

Coronavirus disease 2019 (COVID-19) is a contagious illness worldwide. While guidelines for the treatment of COVID-19 have been established, the understanding of the relationship among neutralizing antibodies, cytokines, and the combined use of antiviral medications, steroid drugs, and convalescent plasma therapy remains limited. Here, we investigated the connection between the immunological response and the efficacy of convalescent plasma therapy in COVID-19 patients with moderate-to-severe pneumonia. The study included a retrospective analysis of 49 patients aged 35 to 57. We conducted clinical assessments to determine antibody levels, biochemical markers, and cytokine levels. Among the patients, 48 (98%) were discharged, while one died. We observed significantly higher levels of anti-nucleocapsid, anti-spike, and neutralizing antibodies on days 3, 7, and 14 after the transfusion compared to before treatment. Serum CRP and D-dimer levels varied significantly across these four time points. Moreover, convalescent plasma therapy demonstrated an immunoregulatory effect on cytokine parameters, with significant differences in IFN-β, IL-6, IL-10, and IFN-α levels observed at different sampling times. Evaluating the cytokine signature, along with standard clinical and laboratory parameters, may help to identify the onset of a cytokine storm in COVID-19 patients and determine the appropriate indication for anti-cytokine treatment.

## 1. Introduction

Coronavirus disease 2019 (COVID-19) is a contagious illness associated with severe acute respiratory syndrome coronavirus 2 (SARS-CoV-2) infection. Patients with this disease display a wide range of physical symptoms, varying from asymptomatic to mild, moderate, and severe, as indicated by several observational studies [1,2,3]. Furthermore, the severity of clinical symptoms is associated with age, ethnicity, gender, and overall health condition [4,5,6,7,8]. Among the many antiviral agents currently available, only a few, such as molnupiravir, paxlovid, and remdesivir [9], have been approved for use in the treatment of COVID-19 [10].

Passive immunization using convalescent plasma from recovered patients (convalescent plasma) containing high levels of neutralizing antibodies is a well-known treatment for a variety of emerging infectious diseases, including those associated with H1N1, SARS-CoV-1, MERS-CoV-1, Ebolavirus, and SARS-CoV-2 [11,12,13,14]. Furthermore, recent evidence suggests that circulating neutralizing antibodies in the serum of recovered COVID-19 patients can effectively inhibit the virus and may have potential benefits for COVID-19 treatment [15]. Neutralizing antibodies are specialized types of immune-system-generated antibodies, crucial for shielding the body against harmful pathogens, including viruses and infectious particles. In a study conducted between January and March 2020, four Chinese patients with COVID-19, including a critically ill pregnant woman, received medical and convalescent plasma therapy and recovered without adverse effects [16]. Additionally, a pilot trial conducted by Duan et al. involving treatment with convalescent plasma with a high antibody titer (>1:640) in 10 patients with severe COVID-19 reported clinical improvement within three days and virus eradication within seven [17]. Another study reported clinical improvements in imaging and laboratory tests among five Chinese participants who underwent COVID-19 convalescent plasma (CCP) therapy [18]. Similarly, Shen et al. [19] reported positive results for five critically ill Chinese patients who received CCP and supportive treatment.

In this study, we explored the connection between the immune response and the effectiveness of combination therapy in a cohort of 62 Thai patients with moderate-to-severe COVID-19. We compared the serum concentrations of anti-SARS-CoV-2 nucleocapsid IgG and anti-spike protein IgG antibodies, as well as neutralizing antibody titers. Additionally, the levels of circulating cytokines, including interleukin-1 beta (IL-1β), IL-6, IL-10, interferon-alpha (IFN-α), IFN-β, and soluble IL-6 subunit (sIL-6R), were measured on the day of transfusion and days 3, 7, and 14 post-transfusion.

## 2. Materials and Methods

### 2.1. COVID-19 Patients

This study aimed to develop and propose a practical approach for utilizing convalescent plasma with compassion to treat patients with severe COVID-19 in Thailand. This retrospective study involved COVID-19 patients hospitalized in various medical centers in Thailand between January and June 2021 whose physicians requested convalescent plasma therapy from the Thai Red Cross Society as a supportive therapy. According to the suggestion for the compassionate use of convalescent plasma of the Thai Red Cross Society, blood samples were collected from recipients on days 0, 3, 7, and 14 post-transfusion to measure the antibody titer and determine whether an additional dose of convalescent plasma should be administered. In addition, questionnaires relating to demographic data and medical history were administered by on-site research nurses or physicians. The inclusion criteria for study participants were: (1) SARS-CoV-2 RNA detected through a nasopharyngeal swab, (2) oxygen saturation ≤ 94%, and (3) evidence of pneumonia, either clinical or radiographic. The criteria for exclusion encompassed the following: (1) patients displaying extreme sensitivity or allergy to plasma or blood products, (2) patients experiencing volume overload, (3) patients who are pregnant or breastfeeding.

The research protocol received approval from the Institutional Review Board (IRB) of the Thai Red Cross National Blood Center (NBC no. 18/2021). The following 10 hospitals in Thailand participated in this study: Chaophraya Hospital (*n* = 1), Ekachai Hospital (*n* = 6), Taksin Hospital (*n* = 7), Thammasat Hospital (*n* = 24), Bangphai Hospital (*n* = 1), Bangpakok Hospital (*n* = 15), Piyavate Hospital (*n* = 1), Phyathai Nawamin Hospital (*n* = 4), Rajavithi Hospital (*n* = 1), and Somdet Phra Pin Klao Hospital (*n* = 2).

### 2.2. Sample Collection

The procedure for collecting blood samples from patients with COVID-19 for antibody testing was as follows: Samples were collected before the transfusion (day 0) and on days 3, 7, and 14 post-transfusion. Blood samples, including serum and plasma, were transported to the Center of Excellence in Clinical Virology and stored at 4 °C. Subsequently, they were aliquoted into 2 mL screw-cap tubes and stored at −80 °C. After the evaluation of antibody titers, the remaining samples were used to test for other biomarkers and cytokines. As all the data collected for analysis in this study were anonymized, the IRB of the Thai Red Cross Society waived the need for written informed consent.

### 2.3. COVID-19 Convalescent Plasma Donors

The Thai Red Cross Society collected all convalescent plasma from COVID-19 donors. The plasma was acquired through apheresis from individuals who had completely recovered from COVID-19 infection (at least 14 days post-recovery). It underwent human leukocyte antigen (HLA) antibody testing to confirm compatibility between the donor and recipient. The plasma contained a minimum neutralizing antibody titer of >1:160 concentration. Furthermore, the plasma underwent an activated pathogen process. Additionally, for the collection of CCP, blood grouping, serological testing for transmitted infections, red cell antigen typing, and direct antiglobulin testing were conducted.

### 2.4. RT-PCR Detection

To detect SARS-CoV-2 RNA, a nasopharyngeal swab was collected from each patient and suspended in a viral transport medium. RNA extraction was performed at the hospital where the patient lived, and the levels of the ORF1a and E genes were quantified. To detect SARS-CoV-2 RNA, the levels of the ORF1a and E genes were quantified using Cobas^®^ SARS-CoV-2 assay. All procedures were conducted following the manufacturer’s instructions.

### 2.5. Antibody Assays

#### 2.5.1. Detection of Anti-SARS-CoV-2 Spike Protein Antibody

The Elecsys anti-SARS-CoV-2 assay (Roche Diagnostics, Basel, Switzerland) was used for the detection of immunoglobulin G (IgG), specific for the receptor-binding domain (RBD) of the SARS-CoV-2 spike protein in serum samples obtained on day 0 (pre-transfusion) and days 3, 7, and 14 (post-transfusion). A concentration below 0.80 U/mL of anti-SARS-CoV-2 spike IgG was considered a negative result, while a concentration equal to or greater than 0.80 U/mL was interpreted as a positive result. The assay has a measurement range of 0.40 to 250 U/mL. For samples with concentrations exceeding 250 U/mL, a 10-fold dilution was applied.

#### 2.5.2. Detection of Anti-SARS-CoV-2 Nucleocapsid Protein Antibody

The Abbott SARS-CoV-2 assay (Abbott Diagnostics, Abbott Park, IL, USA) was employed to detect IgG specific for the nucleocapsid protein of SARS-CoV-2 in serum samples collected on days 0, 3, 7, and 14 following the manufacturer’s guidelines. A result was considered negative for anti-SARS-CoV-2 nucleocapsid antibody when the index (S/C) value was below 1.4 and positive when the index value was equal to or greater than 1.4.

#### 2.5.3. SARS-CoV-2 Neutralizing Antibodies

The Department of Microbiology conducted a live virus microneutralization experiment at the Faculty of Science, Mahidol University, Bangkok, aimed at measuring serum-neutralizing antibody levels using virus neutralization assays. The SARS-CoV-2 virus used in the experiment was obtained from a patient with confirmed COVID-19. For safety reasons, the samples were heat-inactivated by incubation at 56 °C for 30 min. Subsequently, the sera were serially diluted two-fold, starting at a dilution of 1:10. The neutralizing antibody levels were assessed using a standardized dose of 100 (each well of a tissue culture plate contains 100 units of viral concentration) TCID50 (50% tissue culture infectious dose). The neutralizing endpoint was determined by calculating the level of 50% protection against detectable SARS-CoV-2 infection using OD450 nm and OD620 nm signal computation. These techniques followed the methodology outlined by Vacharathit [20].

### 2.6. Cytokine Measurements

COVID-19-positive serum samples were stored at −80 °C. The serum concentrations of the cytokines IL-1β, IL-6, IL-10, IFN-α, IFN-β, and sIL-6R were determined at days 0, 3, 7, and 14. The concentrations of IFN-β and sIL-6R were assessed with the Bio-Plex Pro Human Inflammation Assay, while those of the remaining cytokines were evaluated using the Pro Human Cytokine Screening Panel (Bio-Rad Laboratories, Inc., Hercules, CA, USA). All cytokine concentrations were determined using a Luminex 200 flow cytometer and xPONENT software (Luminex Corporation, Austin, TX, USA). The serum specimens and the reagents used in the analysis were prepared following the manufacturer’s instructions.

### 2.7. Measurement of Serum Biochemical Markers

The diagnostic value of serum biochemical markers (ferritin, C-reactive protein (CRP), D-dimer) was assessed in COVID-19 patients on days 0, 3, 7, and 14. These markers serve as indicators of inflammation and coagulation abnormalities, which are often associated with COVID-19 infection. All of the biochemical markers were sent to Bangpakok 9 International Hospital for processing. The Clinical Biochemistry Laboratory at the Hospital used the Cobas^®^ 6000 modular Biochemistry and Immunoassay analyzer for estimating serum CRP, ferritin, and D-dimer. All reagent kits were from Roche Diagnostics, Basel, Switzerland. The serum CRP estimation was based on a particle-enhanced immunoturbidimetric assay with a <5 mg/L biological reference interval. Serum ferritin estimation used electrochemiluminescence immunoassay (ECLIA) with a reference interval of 30–400 μg/L for males aged 20–60 and 13–150 μg/L for females aged 17–60. Plasma D-dimer estimation relied on a particle-enhanced immunoturbidimetric assay with a <0.5 μg/mL biological reference interval. These reference intervals were provided by the manufacturer in the pack inserts of the diagnostic kits. Quality control assessment used internal quality control material provided by Cobas, Roche Diagnostics, Basel, Switzerland.

### 2.8. Data Analysis

All statistical analyses were performed using GraphPad Prism 9.3.1 (GraphPad, San Diego, CA). Descriptive data were used to analyze participants’ characteristics, and the data were represented using the mean, percentage, and median with the interquartile range (IQR). The chi-square or Fisher’s exact test was used for categorical variables. The differences between groups were examined using the Friedman test in GraphPad Prism 9.3.1. Antibody concentration data were presented using various statistical measures, including the median (range), mean, standard deviation, and percentage. The concentrations of IgG specific to nucleocapsid or spike RBD were reported as sample/cut-off values in U/mL, while the neutralizing antibody concentrations were reported as log10 neutralizing antibody titers. A *p*-value of <0.05 was considered statistically significant. Moreover, in the Appendix A, we provide the inter- and intra-assay CoV, specificity, and sensibility of the kits used for antibody and cytokine evaluations.

## 3. Results

In this study, 176 patients received convalescent plasma treatment and other supportive therapies from November 2020 to May 2021. Of these patients, 9 died, while 167 were discharged and allowed to continue their recovery at home. From January to June 2021, a subset of 62 patients met the eligibility criteria for participating in this study. Of these, 13 were excluded as no samples were collected during the specified periods. The remaining 49 patients provided blood samples for antibody and laboratory analysis. The median age of the participants was 46 years, with an interquartile range of 35 to 57 years. Comorbidities were observed in various patients, with eleven having one additional disease and four having three or more other diseases. Hypertension was the most common comorbidity. Sixteen patients did not have any additional medical issues, while the medical history of the remaining 18 patients was not recorded. Demographic data and the characteristics of the COVID-19 patients are shown in Table 1. All patients received CCP transfusions along with antiviral or steroid medications. The median duration from the onset of symptoms to hospital admission was 8 days, with an interquartile range of 5 to 10 days. The median length of hospital stay for the patients was 17.5 days, with an interquartile range of 17 to 22.5 days (Table 1). After 30 days, one patient passed away, while the remaining 48 were discharged and continued their recovery at home.

### 3.1. The Anti-Nucleocapsid Protein IgG Antibody Response in Patients with COVID-19

This study assessed the seroconversion trend in 49 patients with moderate-to-severe COVID-19. To assess the antibody response, we measured the levels of IgG antibodies specific for two primary SARS-CoV-2 structural proteins—spike (RBD) and nucleocapsid proteins. The measurements were taken multiple times, namely, on day 0 (before convalescent plasma transfusion) and on days 3, 7, and 14 after CCP transfusion. We found that the anti-nucleocapsid antibody level was significantly lower on day 0 than on days 3, 7, and 14 post-transfusion (Figure 1A). No difference in anti-nucleocapsid protein IgG levels was observed among patients who had been ill for different durations (0–3, 4–7, 8–11, and 12–15 days) (Figure 1B).

### 3.2. The Anti-Spike RBD Protein IgG Antibody Response in Patients with COVID-19

The anti-spike receptor-binding domain (RBD) levels gradually increased from day 0 to days 3, 7, and 14 post-transfusion (Figure 2A). However, there were no significant differences in the anti-spike RBD protein IgG levels observed among patients who had been ill for different durations (0–3, 4–7, 8–11, and 12–15 days) (Figure 2B).

### 3.3. The Neutralizing Antibody Titer in Patients with COVID-19

Neutralizing antibody titers significantly increased from day 0 to days 3, 7, and 14 after CCP transfusion (Figure 3A). However, there were no significant differences in neutralizing antibody levels among patients who had been ill for different durations (0–3, 4–7, 8–11, and 12–15 days) (Figure 3B).

### 3.4. The Levels of Laboratory Markers and Circulating Cytokines

In the study, we analyzed the concentrations of various biomarkers and cytokines in patients with COVID-19 to determine the changes occurring over time and in response to CCP transfusion. The results showed significant differences in CRP and D-dimer levels at different time points. The CRP levels were significantly higher on the day before CCP transfusion than on days 3 and 7 after transfusion but were moderately higher on day 14 than on day 7. Although the D-dimer levels also showed significant differences among the different time points, the specific relationship between the levels was not clear. Furthermore, the ferritin levels were high at all four time points, with no differences detected among the sampling times.

In addition to the biochemical markers, we also analyzed the levels of several cytokines, namely, IL-1β, IL-6, IL-10, IFNα2, IFN-β, and soluble IL-6 receptor subunit alpha (sIL-6R). At various sampling times, there were noticeable differences in the IFN-β, IL-10, IFNα2, and IL-6 quantities. Specifically, the levels of IFN-β and IL-10 exhibited significant differences (*p* = 0.0168CV and <0.0001*, respectively). At the same time, IFNα2 and IL-6 levels also showed significant differences (*p* < 0.0005* and <0.0001*, respectively) among all the groups, although the changes were relatively minor. Interestingly, IL-6 levels were higher on day 0 than on days 3 and 7 but were higher on day 14 than on day 7. No differences in the levels of sIL-6RB and IL-1β were observed among the different time points assessed (Table 2). The cytokine levels of individual COVID-19 patients, represented for the days before COVID-19 convalescent plasma (CCP) transfusion as well as 3, 7, and 14 days after transfusion, are shown in Figure 4.

## 4. Discussion

This study included 49 patients with moderate-to-severe COVID-19 from 10 hospitals in Thailand. The most common comorbidities observed were hypertension and diabetes. Half of the participants had comorbidities, in line with a previous study conducted in Wuhan, China [3].

The results of our retrospective investigation indicated that high antibody titers (>1:160) in recovered COVID-19 patients are associated with improved clinical outcomes. Furthermore, the combination of CCP, antiviral medication, and steroids yields satisfactory results in COVID-19 patients with moderate-to-severe disease, as categorized by the World Health Organization.

Regrettably, one participant in our study passed away in the hospital, while the remaining patients were discharged. Our findings agree with those of a previous study indicating that the early administration of convalescent plasma with high antibody titers in the early stages of COVID-19 leads to better outcomes in patients with moderate and severe symptoms [21]. Similarly, a study by Shen et al. (2020) demonstrated that the clinical symptoms of COVID-19 patients improve after receiving convalescent plasma containing neutralizing antibodies [19].

Our and previous results suggest that COVID-19 patients with moderate-to-severe symptoms who receive convalescent plasma units with high antibody titers in the early phases of the disease exhibit a high rate of improvement. These findings provide further support for the potential effectiveness of convalescent plasma therapy in managing COVID-19, particularly among patients with more severe disease.

CRP is associated with inflammation in disease progression and poor clinical outcomes [22]. Our investigation showed that CRP levels in the median range were significantly higher before CCP transfusion than on days 3 and 7 after transfusion but were higher on day 14 than on day 7. This suggests that CCP therapy may reduce CRP levels in patients with COVID-19. D-dimer, a protein involved in blood clotting and fibrogenesis, has been associated with severe COVID-19 [23]. In our study, the D-dimer levels exhibited substantial variation across time points, although the correlation values between sampling times were unclear. Ferritin, a modulator of immune dysregulation, has both immunosuppressive and proinflammatory activities and plays a role in cytokine storms [24]. One study showed that, in COVID-19 patients, ferritin levels are linked to disease severity and can serve as a prognostic risk factor [25]. In our study, ferritin levels remained high at all four time points assessed, with no differences observed among the sampling times.

To evaluate the effect of antibody levels on the response to convalescent plasma therapy and clinical outcomes, we measured the serum concentrations of neutralizing antibodies, anti-SARS-CoV-2 nucleocapsid IgG antibodies, and anti-SARS-CoV-2 spike RBD IgG antibodies. Our findings demonstrated that serum levels of anti-nucleocapsid IgG and anti-spike protein IgG antibodies were significantly lower before transfusion than on days 3, 7, and 14 after transfusion. Neutralizing antibody concentrations showed a similar trend to the levels of IgG antibodies against both SARS-CoV-2 proteins. Notably, in the case of COVID-19-related death, the levels of IgG antibodies against the two viral proteins were found to be low before transfusion, consistent with earlier research that associated delayed seroconversion with poor virus control and higher morbidity [26]. In addition, low levels of anti-SARS-CoV-2 spike protein IgG antibodies have been correlated with increased mortality [27]. COVID-19 patients with critically severe disease who require ventilatory and extracorporeal membrane oxygenation (ECMO) support exhibit the highest SARS-CoV-2 antibody titers [28]. Our data showed that CCP therapy may be less effective in COVID-19 patients displaying low IgG antibodies against nucleocapsid and spike proteins but high neutralizing antibodies. However, given that only one patient died in our study, we could not statistically analyze the association among IgG levels, neutralizing antibody concentrations, and mortality rates. It is essential to consider that factors other than antibody levels may also play a significant role in treatment outcomes.

Like antibodies, cytokines are also critical components of the immune response and can serve as prognostic markers in severe disease [29,30]. Aberrant levels of various cytokines and chemokines have been observed in COVID-19 patients [31,32,33]. Here, we also evaluated the immunoregulatory effect of CCP treatment on cytokine storm parameters, including the concentrations of IL-1β, IL-6, IL-10, IFN-α, IFN-β, and sIL-6Rα. However, none of these cytokines showed a specific pattern at any time point assessed.

IL-6 can influence the immunological response and hematopoiesis [34] by inducing the production of acute-phase proteins such as CRP and fibrinogen during inflammation [35]. Meanwhile, IL-10 is a predominant cytokine during influenza infection and plays a role in stimulating the adaptive immune system [36]. In our study, the levels of the cytokines IL-6 and IL-10 differed significantly before and after transfusion, which is consistent with earlier studies linking these cytokines to poor prognosis and critical illness [32,37,38].

Type I interferons, which include IFN-α and IFN-β, comprise a family of cytokines with a vital function in antiviral responses and have a complex role in COVID-19. Studies have reported varying IFN-α responses in mild and moderate-to-severe cases [39]. Here, we found significant differences in IFN-α and IFN-β levels before and after transfusion; however, the exact profile of these differences remains uncertain. Regarding IL-1β and sIL-6R, we did not find significant alterations in their levels before and after transfusion. It is important to note that various factors, including the severity of the disease, individual immune responses, viral load, comorbidities, genetic mutation, ethnicity, and the timing of treatment interventions, can influence cytokine levels in COVID-19 patients. Consequently, the cytokine levels might have altered during the mentioned time periods.

While our study provides valuable insights into the effectiveness of convalescent plasma therapy in COVID-19 patients, it also had some limitations. Specific data, including data on comorbidities, were unavailable for some patients owing to challenges faced during the pandemic in Thailand. Additionally, blood collection for testing was not possible for all patients. Another limitation was the concurrent administration of other treatments, such as antiviral and steroid drugs, alongside convalescent plasma, which could have influenced the results. Thus, it is impossible to draw a definitive conclusion regarding the association among cytokine storm, IgG antibody levels, and neutralizing antibody concentrations in patients with COVID-19. Nevertheless, our data suggest that convalescent plasma with high antibody titers can be effective and safe in treating COVID-19, and combining it with other antiviral or steroid medications may improve the survival rate of patients.

## Figures and Tables

**Figure 1 diseases-11-00112-f001:**
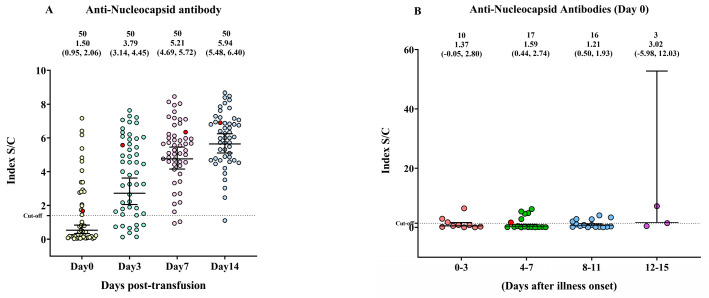
The anti-nucleocapsid protein immunoglobulin G (IgG) response in patients with COVID-19 on the day before COVID-19 convalescent plasma (CCP) transfusion (day 0) and on days 3, 7, and 14 post-transfusion. (**A**) The changes in anti-nucleocapsid protein IgG antibody on days 0, 3, 7, and 14. (**B**) The levels of IgG antibodies against the anti-nucleocapsid protein were compared across four time periods after the onset of illness. The anti-nucleocapsid IgG S/C index distribution is depicted by a scattered dot plot with a median and interquartile range (IQR). A red circle symbolizes death.

**Figure 2 diseases-11-00112-f002:**
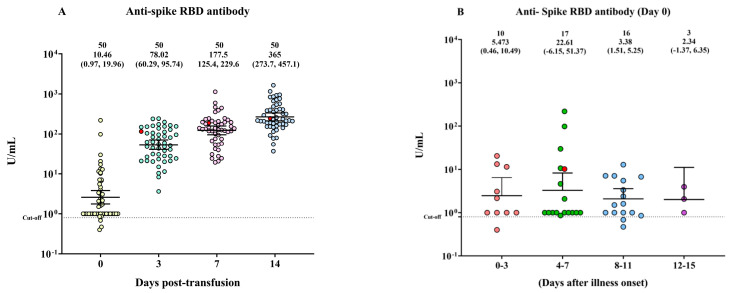
The anti−spike receptor−binding domain (RBD) IgG response in patients with COVID-19 on the day before COVID-19 convalescent plasma (CCP) transfusion (day 0) and on days 3, 7, and 14 post-transfusion. (**A**) The changes in anti−spike RBD IgG levels on days 0, 3, 7, and 14. (**B**) The levels of IgG antibodies against the anti−spike RBD protein were compared across four time periods after the onset of illness. The distribution of anti-spike IgG (U/mL) is depicted by a scattered dot plot with a median and interquartile range (IQR). A red circle symbolizes death.

**Figure 3 diseases-11-00112-f003:**
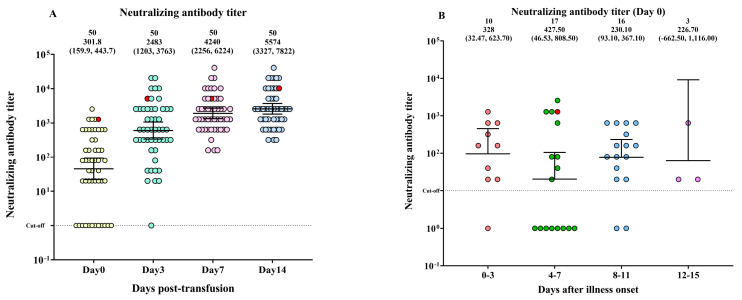
The neutralizing antibody titer in patients with COVID-19 on the day before COVID-19 convalescent plasma (CCP) transfusion (day 0) and on days 3, 7, and 14 post-transfusion. (**A**) Neutralizing antibody titer trends on days 0, 3, 7, and 14. (**B**) The levels of neutralizing antibodies were compared across four time periods after the onset of illness. The distribution of log_10_ neutralizing antibody titer is depicted by a scattered dot plot with a median and interquartile range (IQR). A red circle symbolizes death.

**Figure 4 diseases-11-00112-f004:**
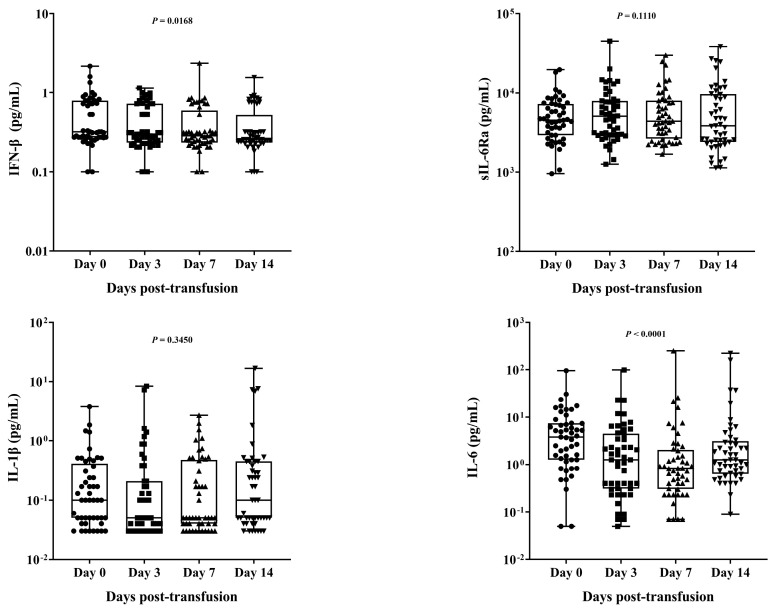
The measurement range of cytokines was evaluated in all samples collected from severe COVID-19 patients (n = 49) who had undergone convalescent plasma transfusion (CCP) on the day before the treatment in contrast to serum samples obtained on days 3, 7, and 14 after the treatment. The summary statistics illustrated in the box plot encompass the subsequent features: Each data point is represented by a dot, the centerline indicates the median, and the lower and upper hinges correspond to the first and third quartiles, respectively, while the upper and lower whiskers extend from these hinges to encompass the highest and lowest values, respectively. The distinctions between these groups were assessed through the utilization of the Friedman test. The comparisons derived from the Friedman test revealed significantly disparate levels of IFN−β (*p* = 0.0168), IL−6 (*p* < 0.0001), IL−10 (*p* < 0.0001), and IFNα2 (*p* < 0.0005) in samples taken on day 0 when compared to samples obtained on days 3, 7, and 14.

**Table 1 diseases-11-00112-t001:** Demographic data of the study population and serum biochemical characteristics on admission to hospital.

Characteristics	COVID-19 Patients (*N* = 49)
Age, mean (range) years	46 (35–57)
Sex, female/male, n	18/31
Comorbidities, n = 16	
Diabetes mellitus	6
Hypertension	6
Dyslipidemia	3
Obesity	6
Gout	1
Chronic kidney disease	1
Fatty liver disease	1
No underlying disease, n	16
No data available, n	18
The median (IQR) days from the onset of symptoms to hospital admission	8 (5–10)
Median (IQR) days of hospitalization	17.5 (17–22.5)
Clinical outcomes (survival, %)	97.96%

**Table 2 diseases-11-00112-t002:** The levels of biochemical markers and circulating cytokines pre- and post-transfusion.

Biochemical Marker	Pre-Transfusion	Post-Transfusion			*p*-Value
	Day 0	Day 3	Day 7	Day 14	
C-reactive protein (mg/L), normal range (0–6)					
Median	31.610	9.935	2.065	5.945	
Range	(0.6–244.2)	(0.6–130.64)	(0.6–201.26)	(0.6–222.21)	<0.0001 *
D-dimer (µg/mL)					
Median	0.1900	0.2850	0.4550	0.2550	
Range	(0.15–3.28)	(0.15–9.68)	(0.15–9.72)	(0.15–9.15)	0.0008 *
Ferritin (ng/mL), normal range (13–400), age and gender dependent					
Median	729.3	734.3	769.5	745.5	
Range	(43.2–3021)	(26.3–3227.7)	(16.5–3137.7)	(28.1–3244.6)	0.8896
Circulating cytokine					
Interferon beta (IFN-β) (pg/mL)					
Median	0.32	0.30	0.28	0.26	
(Range)	(0.22–2.17)	(0.20–1.15)	(0.18–2.37)	(0.18–1.56)	
Mean ± SD	0.53 ± 0.40	0.44 ± 0.28	0.42 ± 0.36	0.40 ± 0.30	0.0168 *
Soluble interleukin 6 receptor alpha (sIL-6R)					
(pg/mL)					
Median	4500	5076	4382	3849	
(Range)	(954.61–19614.51)	(1255.99–44828.6)	(1680.57–29896.14)	(1127.57–38273.86)	
Mean ± SD	5507 ± 3702	6847 ± 6936	6592 ± 5957	7246 ± 7874	0.1110
Interleukin-1 beta (IL-1β)					
Median	0.10	0.05	0.05	0.10	
Range	(0.03–3.80)	(0.03–8.37)	(0.03–2.72)	(0.03–16.76)	
Mean ± SD	0.34 ± 0.64	0.53 ± 1.57	0.31 ± 0.54	0.99 ± 2.88	0.3450
Interleukin-6 (IL-6)					
Median	3.80	1.26	0.82	1.26	
Range	(0.05–95.66)	(0.05–98.82)	(0.07–248.72)	(0.09–222.37)	
Mean ± SD	7.648 ± 14.39	5.138 ± 14.54	7.563 ± 35.55	11.37 ± 38.70	<0.0001 *
Interleukin-10 (IL-10)					
Median	1.38	0.36	0.19	0.36	
Range	(0.03–116.37)	(0.03–69.68)	(0.03–38.09)	(0.03–21.24)	
Mean ± SD	4.263 ± 16.43	2.387 ± 9.912	1.352 ± 5.419	1.043 ± 3.111	<0.0001 *
Interferon-alpha 2 (IFNα2)					
Median	0.44	0.40	0.39	0.39	
Range	(0.29–5.43)	(0.29–0.54)	(0.25–0.61)	(0.29–0.54)	
Mean ± SD	0.7580 ± 1.099	0.41 ± 0.06	0.41 ± 0.07	0.40 ± 0.05	0.0005 *

The levels of biochemical markers and circulating cytokines were measured in COVID-19 patients on the day prior to COVID-19 convalescent plasma (CCP) transfusion (day 0) and on days 3, 7, and 14 after the transfusion. The differences between these groups were assessed using the Friedman test. A significant difference between the groups is indicated by *, *p*-value <0.05.

## Data Availability

The datasets generated and analyzed during the current study are available from the corresponding author upon reasonable request.

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
