# Peer review of "Dynamics of Cytokine, SARS-CoV-2-Specific IgG, and Neutralizing Antibody Levels in COVID-19 Patients Treated with Convalescent Plasma"

_diseases, 2023, doi:10.3390/diseases11030112_

Round 1

Reviewer 1 Report

For COVID-19 have been emerged more than three years, medicine and vaccine have been widely used, tconvalescent plasma therapy have not been suitable for COVID-19 tratment. herefore, the scientific meanings of this study is limited.

The patients number described in results section and discussion section was not consisten. Some conclusions were over-stated.

Author Response

Dear Reviewer 1,

Subject: Submission of the Revised Manuscript. Manuscript ID: diseases-2554299.

We greatly appreciate your feedback. After a thorough review of your comments, we have diligently revised the manuscript as per your suggestions. Our responses are provided in a point-by-point format and are enclosed as a PDF file (Please see the attachment). Noteworthy alterations in the manuscript are highlighted in bold red font.

We trust that this revised version aligns well with publication standards, and we eagerly anticipate your response in the near future.

Best regards,

Pornpitra Pratedrat, Ph.D.

Reviewer 2 Report

Diseases: Dynamics of cytokine, SARS-CoV-2-specific IgG, and neutralizing antibody levels in COVID-19 patients treated with convalescent plasma

In this manuscript, Pratedrat et al. determine antibody, biological marker, and cytokine levels in COVID-19 patients who received convalescent plasma from recovered COVID-19 patients (retrospective study). They determined an increase in antibody levels than before receiving the plasma (expected), and some differences in some biomarker and cytokine levels.

General comment: It is expected that antibody levels (all types measured) would increase after receiving the convalescent plasma, since that is the point of receiving the plasma. The focus of these data should be more focused on the 7-14 days where the antibodies remain/are not going down, etc.

Specific comments:

Section 2.4. This section needs to be expanded. What kit was used (since referenced to manufacture’s instructions)? What were the primer sequences/RT-PCR conditions?

2.7. The end of this section needs to be expanded (so a reader can perform the assay). What was the procedure for processing (and how was each of the markers read on the machine)?

Section 3: The authors state when there is a “significant” result, but no statistics are included in the applicable figures/figure legends, including what type of statistical test was used (the methods state GraphPad Prism was used to do the calculations). Same with Table 1-what statistical tests were used?

Table 1: The authors use an asterisk (*) to indicate that IFN-beta levels are significantly different, but what is defined as significant in the table legend (p<0.0005) would make the interferon-beta levels not significant. Do you mean p< 0.05 is significant? All of the means/standard deviations highly overlap, so without knowing what type of test was used (Sec. 3 comment above) this is hard to believe a p value of <0.05 would be accomplished.

Author Response

Dear Reviewer 2,

Subject: Submission of the Revised Manuscript. Manuscript ID: diseases-2554299.

We greatly appreciate your feedback. After a thorough review of your comments, we have diligently revised the manuscript as per your suggestions. Our responses are provided in a point-by-point format and are enclosed as a PDF file (Please see the attachment). Noteworthy alterations in the manuscript are highlighted in bold red font.

We trust that this revised version aligns well with publication standards, and we eagerly anticipate your response in the near future.

Best regards,

Pornpitra Pratedrat, Ph.D.

Reviewer 3 Report

The paper by Pratedrat and colleagues wants to show the relevance of convalscent plasma therapy in COVID-19 moderate-to-severe patients.

Despite the paper is well-written, major essential issues need to be addressed.

Please, provide a definition of "neutralizing antibodies" in the Introduction.

The authors state that they "...explored the connection...". Connected events should need to be experimentally demonstrated, so I suggest to replace "connection". You showed how the convalescence serum therapy can likely affect other parameters (with several limitations).

The first big concern is the number of patients with no specific clinical data: although the authors assumed this as a limitation, 27 are the most of patients included in this study. How can we exclude the impact of other clinical causes? Subsequently, how can we consider valid the presented data?

Which day after the disease onset the CCP was administered? And when it was collected (day after the COVID-19 recovery)?

Exclusion criteria for study participants: particularly, was the presence of autoimmune diseases excluded? Again, the available data are poor.

RT-PCR for virus detection: "All procedures were conducted following the manufacturer's instructions". Can the authors indicate the kit used and include this information within the text?

Detection of Anti-SARS-CoV-2 Spike Protein Antibody: "For samples with concentrations exceeding 250 U/mL, a 10-fold dilution was applied". To my knowledge, the samples should be tested all in the same experimental conditions, otherwise this procedure may result in a bias.

SARS-CoV-2 Neutralizing Antibodies: "The neutralizing antibody levels were assessed using a standardized dose of 100 TCID50 (50% tissue culture infectious dose). The neutralizing endpoint was determined by calculating the level of 50% protection against detectable SARS-CoV-2 infection using OD450 nm and OD620 nm signal computation". Even though already published by Vacharathit et al., can the authors report some specifics of the protocol used (for example, 100 stands for?)?

Data Analysis: "The concentrations of IgG specific for nucleocapsid or spike RBD were reported as sample/cut-off values in U/mL". Could the authors explain the need to apply the index (as it is defined by the authors) sample/cut-off? Knowing the cut-off value, could only the sample concentration value be reported?

Results: "Comorbidities were observed in various patients, with six having one other disease (WHICH ONE?), and three having three or more other diseases (SEE THE NOTE ABOVE). Hypertension was the most common comorbidity. Sixteen patients did not have any additional medical issues, while the medical history of the remaining 27 patients was not recorded (as already highlighted, this is a crucial point). Missing information could be key information.

In Table 1, I suggest to include the number of male patients (in this way: Sex, Female/Male, n. 18/31), deleting the percentage and to report all the ranges as 35-70 (not 35, 70. Please, apply to all).

The Anti-Nucleocapsid Protein IgG Antibody Response in Patients with COVID-19: "No difference in anti-nucleocapsid protein IgG levels was observed among patients who had been ill for different durations (0–3, 4–7, 8–11, and 12–15 days). I apologize, I didn't understand. Could this be clearly explained? The same for "...no significant differences in the levels of anti-spike RBD protein IgG observed among patients who had been ill for different durations..." and for "...are no significant differences in neutralizing antibody levels among patients who have been ill for different durations...".

In the period of time considered: the levels of D-dimer significantly increased; the ferritin concentration shows no significant changes, and so is also for IL-6, which is considered one of the most important biomarker of COVID-19 severity, the first studied at the onset of the pandemic. All these aspects have to be more better discussed and explained, including the fact that at day 14, the changing parameters seem to increase again.

In my opinion a graphical representation of cytokine levels could be useful and appropriate.

Discussion: "Meanwhile, IL-10 is a predominant cytokine during influenza infection and plays a role in stimulating the adaptive immune system": IL-10 has a well-know anti-inflammatory role. In this study, IL-10 levels decrease at the time points observed. Can the authors give a reason for this?

In Figure 1-2-3, a sequence of numbers is reported on the top of the graphs. Are they the number of patients, the median and interquartile range? Moreover, a red dot can be found in the graph. Please, specify and explain in the figure legends. In addition, the term "kinetic" is reported referring to the trend/change of the antibody levels. Since this term has a specific scientific meaning, I suggest to replace it. Lastly, in Figure 1 on the y axis "Index S/C" (or maybe S/C index?) is reported, while in Figure 2 U/mL can be observed. Why are they different? M&M reported the same unit of measurement for anti-spike and anti-nucleocapsid antibodies.

Please, include inter- and intra-assay CoV, specificity and sensibility of the kits used for antibody and cytokine evaluations.

In Table 2, asterisk (*) and b (b) as single quote marks are visible, but in my opinion they are typos. Please, correct.

Lines 270-271: "Significant differences in the levels of IFN-β, IL-10, IFNα2, and IL-6 among the different sampling times". This is an incomplete sentence.

Lines 280-281: "The average age of the patients was 46 years and 63.3% of the participants were male". I would delete this sentence and include this information in Table 1, as previously suggested.

Lines 326: "ECMO"stands for?

Author Response

Dear Reviewer 3,

Subject: Submission of the Revised Manuscript. Manuscript ID: diseases-2554299.

We greatly appreciate your feedback. After a thorough review of your comments, we have diligently revised the manuscript as per your suggestions. Our responses are provided in a point-by-point format and are enclosed as a PDF file (Please see the attachment). Noteworthy alterations in the manuscript are highlighted in bold red font.

We trust that this revised version aligns well with publication standards, and we eagerly anticipate your response in the near future.

Best regards,

Pornpitra Pratedrat, Ph.D.

Reviewer 4 Report

The manuscript shows the therapeutic effect of treatment of moderate-to-severely ill COVID-19 patients treated with convalescent plasma with proven neutralizing antibody titer  (>1:160). A significant increase in anti-nucleocapsid, anti-Spike RBD and neutralizing antibodies was found, together with significant impact on the levels of CRP, D-dimer and several cytokines, in particular IL-6 and IL-10. A favorable clincial outcome (only 1 died out of 48 (62)) was discussed. The discussion should, however, be supported by more data from literature regarding survival/recovery rates in patients of similar severity. The manuscript is clearly written and the data are well presented. A major limitation of this study are the various co-treatments, including antiviral and steroid therapy, which can be assumed to significantly affect the clicniacal outcome and lab parameters, however, no correlation to these data of cotreatments is provided. The authors are encouraged to show such data if available. Also the antibody titers (i.e. kinetics before vs. after treatment) and the blood parameters (kinetics) of the patient who died may be presented in comparison to all patients who recovered. This may lead to the identification of clinically important blood parameters 

teh manuscript is well written

Author Response

Dear Reviewer 4,

Subject: Submission of the Revised Manuscript. Manuscript ID: diseases-2554299.

We greatly appreciate your feedback. After a thorough review of your comments, we have diligently revised the manuscript as per your suggestions. Our responses are provided in a point-by-point format and are enclosed as a PDF file (Please see the attachment). Noteworthy alterations in the manuscript are highlighted in bold red font.

We trust that this revised version aligns well with publication standards, and we eagerly anticipate your response in the near future.

Best regards,

Pornpitra Pratedrat, Ph.D.

Round 2

Reviewer 1 Report

I am satisfied with the revisions. I agree with the publication of this manucript in Diseases.

Author Response

August 24, 2023

Dear Reviewer 1,

Resubmission of Manuscript Reference No. ID: diseases-2554299 (round 2)

Enclosed herewith is the revised version of our manuscript, originally titled "Dynamics of Cytokine, SARS-CoV-2-Specific IgG, and Neutralizing Antibody Levels in COVID-19 Patients Treated with Convalescent Plasma." We are submitting this version for your consideration for publication in Diseases MDPI. We have included a point-by-point response to the reviewer's comments, and the revised sections of the manuscript are highlighted in red font with underlining.

We are confident that the revisions we have made to the manuscript, along with our detailed responses, address the reviewer's feedback adequately, making our manuscript suitable for publication in Diseases MDPI. We eagerly anticipate your response in due course.

Sincerely,

Pornpitra Pratedrat, Ph.D.

Reviewer 2 Report

While the authors have adequately addressed reviewer comments in their response to reviewers, they are not included in the updated manuscript (for example-manufacturer of the kits used and the statistical tests need to be given to the readers, not just the reviewers). 

Author Response

August 24, 2023

Dear Reviewer 2,

Re: Resubmission of Manuscript Reference No. ID: diseases-2554299 (round 2)

Enclosed herewith is the revised version of our manuscript, originally titled "Dynamics of Cytokine, SARS-CoV-2-Specific IgG, and Neutralizing Antibody Levels in COVID-19 Patients Treated with Convalescent Plasma." We are submitting this version for your consideration for publication in Diseases MDPI. We have included a point-by-point response to the reviewer's comments, and the revised sections of the manuscript are highlighted in red font with underlining.

We are confident that the revisions we have made to the manuscript, along with our detailed responses, address the reviewer's feedback adequately, making our manuscript suitable for publication in Diseases MDPI. We eagerly anticipate your response in due course.

Sincerely,

Pornpitra Pratedrat, Ph.D.

Reviewer 3 Report

The Authors addressed only some raised concerns.

All the explanations requested have to be included in the main text of the manuscript. In particular:

1. Include the definition of "neutralizing antibodies" in the Introduction.

3.  The Authors didn't answered: when was the CCP collected from patients, in order to save and use it as COVID-19 therapy? (day after the COVID-19 recovery).

4. Include the exclusion criteria for study participants in the appropriate section of M&M.

5. Include the name of the commercial kit used for RT-PCR in the appropriate section of M&M.

6. I perfectly know the ELISA protocol and I insist on the need to use the same dilution for all the samples to be tested, since adjust the dilution on a case by case basis can result artificial. A preliminary assay has to be performed, in order to choose the exact dilution to be used for all the samples. In this way, the experiment will turn out to be accurate. The samples having concentration within the range of the assay will result positive or negative in any case, even if diluted (furthermore, the sample concentration obtained will be multiplied by the dilution factor).

7. Include the provided response and some other details (as requested) of the protocol used in the M&M section.

8. The S/C is an index, so S/C and U/ml are numerically different. In my opinion, it is not possible the use of the same scale number (y-axis) in the graphs (see figures 1 and 2). I suggest to not use the ticks on the y-axis of the graphs.

10. I recommend to put the dash (not the comma) between the two numbers of a range in Tables 1 and 2.

11. Include this explanation in the appropriate point of the main text.

12. I suggest to include this clarification in the Discussion.

13. I don't see what the presence of comorbidities has to do with the graphical representation of cytokine levels. In addition, it is normal to observe fluctuating cytokine levels in patients (standard deviations depend on this aspect). A box & whisker plot can represent the level of every cytokine in all the patients.

16. In a paper describing cytokine, SARS-CoV-2-specific IgG, and neutralizing antibody levels in COVID-19 patients, I suggest to add the requested info in as Supplementary material at least.

Author Response

August 24, 2023

Dear Reviewer 3,

Re: Resubmission of Manuscript Reference No. ID: diseases-2554299 (round 2)

Enclosed herewith is the revised version of our manuscript, originally titled "Dynamics of Cytokine, SARS-CoV-2-Specific IgG, and Neutralizing Antibody Levels in COVID-19 Patients Treated with Convalescent Plasma." We are submitting this version for your consideration for publication in Diseases MDPI. We have included a point-by-point response to the reviewer's comments, and the revised sections of the manuscript are highlighted in red font with underlining.

We are confident that the revisions we have made to the manuscript, along with our detailed responses, address the reviewer's feedback adequately, making our manuscript suitable for publication in Diseases MDPI. We eagerly anticipate your response in due course.

Sincerely,

Pornpitra Pratedrat, Ph.D.

Reviewer 4 Report

The authors tried to address the reviewers comments. The manuscript can be recommended for publication

Author Response

August 24, 2023

Dear Reviewer 4,

Re: Resubmission of Manuscript Reference No. ID: diseases-2554299 (round 2)

Enclosed herewith is the revised version of our manuscript, originally titled "Dynamics of Cytokine, SARS-CoV-2-Specific IgG, and Neutralizing Antibody Levels in COVID-19 Patients Treated with Convalescent Plasma." We are submitting this version for your consideration for publication in Diseases MDPI. We have included a point-by-point response to the reviewer's comments, and the revised sections of the manuscript are highlighted in red font with underlining.

We are confident that the revisions we have made to the manuscript, along with our detailed responses, address the reviewer's feedback adequately, making our manuscript suitable for publication in Diseases MDPI. We eagerly anticipate your response in due course.

Sincerely,

Pornpitra Pratedrat, Ph.D.

Round 3

Reviewer 3 Report

The revisions have improved the scientific soundness of the paper, which is now more complete.

As already indicated, in Figures 2 and 3 I suggest to eliminate the ticks between the numbers of the scale. Furthermore, I also suggest to uniform the scale of the graphs: in Figure 2, the y-axis is up to 104 in the left graph and 103 in the right graph; in Figure 3, the y-axis in the graph on the left is up to 105, while in the graph on the right is up to 104. This suggestion may help to notice the differences.

As already indicated for Figures 1 and 2 (Point 8 of the rebuttal letter), is it possible that y-axes also in Figures 2 and 3 have the same scale? Log10 and U/mL are numerically different. Is there an explanation?

Regarding the cytokine levels, the statistical significances included in Table 1 and in the graphs in Figure 4 are related to.....what? Please, specify in the figure legend, in the table and within the text. Furthermore, if at the first mention you report the statistical significance as 0.0168, then it would be correct not to round up to 0.017 (line 303).

Author Response

August 27, 2023

Dear Reviewer,

Resubmission of Manuscript Reference No. ID: diseases-2554299

Enclosed herewith is the revised version of our manuscript, originally titled "Dynamics of Cytokine, SARS-CoV-2-Specific IgG, and Neutralizing Antibody Levels in COVID-19 Patients Treated with Convalescent Plasma." We are submitting this version for your consideration for publication in Diseases MDPI. We have included a point-by-point response to the reviewer's comments, and the revised sections of the manuscript are highlighted in red font with yellow highlight.

We are confident that the revisions we have made to the manuscript, along with our detailed responses, address the reviewer's feedback adequately, making our manuscript suitable for publication in Diseases MDPI. We eagerly anticipate your response in due course.
